# OpenReview forum: "Cheating Automatic LLM Benchmarks: Null Models Achieve High Win Rates"
_NeurIPS.cc/2024/Workshop/SafeGenAi — SafeGenAi Poster_

### Official Review · Reviewer_2xY9 · 2024-10-08
**Interesting proof-of-concept with limited applicability to real-world scenarios (yet)**

**Rating:** 6
**Confidence:** 3

**Review:**

The authors highlight the limitations of existing benchmarks by demonstrating that generic persuasive responses can obtain high win rates in state-of-the-art evaluations.

Strengths:
* Paper is very clear and experimental setup is sound.
* The authors conduct many different ablations to understand the impact on their setup on a wide range of evaluators and benchmarks.
* The context is well-motivated and most relevant limitations acknowledged.

Limitations:
* To me, the main limitation of the findings are their applicability to real-world scenarios. Most of the results are equivalent to showing that you can create an adversarial suffix that gets a model to output a specific response (the selected model). In a real-scenario, an adversary may want to fool a judge while providing responses that look similar to benign responses and preserve utility while being "stealthy". I encourage authors to explore ways of incorporating these findings into a model that can be optimized to fool automated judges during training.

Suggestions:
* I think the authors may consider including an example of what they mean by "constant outputs" early on so that the reader gets an early intuition of how these placeholders look like. Maybe just bring Figure 1 to the second page.

---

### Official Review · Reviewer_HFCK · 2024-10-09
**Interesting paper highlighting weaknesses in LLM auto-evals.**

**Rating:** 9
**Confidence:** 4

**Review:**

This paper presents an argument that LLM auto-eval systems are prone to cheating via adversarially-designed prompts. To test this hypothesis, the authors craft prompts that achieve "state-of-the-art" win-rates on some existing benchmarks despite being trivially wrong, i.e. the same, crafted response for each given prompt.

The paper is well-motivated, clearly written, and provides a compelling argument that cheating is indeed not just possible but relatively straightforward on auto-evals. The background shows that the authors have a good understanding of the field.

One weakness with the paper is the lack of detail of their adversarial random search method to generate these prompts. Relatively little detail is provided other than a short algorithm in the appendix. I'm not fully convinced that this algorithm works that well; Figure 2 shows that indeed, there are only exactly 3 steps where the loss drops at all. I would appreciate some more detail here beyond some citations of previous works. However, I believe that it is more important that such a prompt was found at all than the details of how it was found, so I don't think it's a significant problem.